# RESHAPING ACTIVATION FUNCTIONS: A FRAMEWORK FOR ACTIVATION FUNCTION OPTIMIZATION BASED ON MOLLIFICATION THEORY

## ABSTRACT

The deep learning paradigm is progressively shifting from non-smooth activation functions, exemplified by ReLU, to smoother alternatives such as GELU and SiLU. This transition is motivated by the fact that non-differentiability introduces challenges for gradient-based optimization, while an expanding body of research demonstrates that smooth activations yield superior convergence, improved generalization, and enhanced training stability. A central challenge, however, is how to systematically transform widely used non-smooth functions into smooth counterparts that preserve their proven representational strengths while improving differentiability and computational efficiency. To address this, we propose a general activation smoothing framework grounded in mollification theory. Leveraging the Epanechnikov kernel, the framework achieves statistical optimality and computational tractability, thereby combining theoretical rigor with practical utility. Within this framework, we introduce Smoothed ReLU (S-ReLU), a novel second-order continuously differentiable ($C^2$) activation derived from ReLU that inherits its favorable properties while mitigating inherent drawbacks. Extensive experiments on CIFAR-10, CIFAR-100, and ImageNet-1K with Vision Transformers and ConvNeXt consistently demonstrate the superior performance of S-ReLU over existing ReLU variants. Beyond computer vision, large-scale fine-tuning experiments on language models further show that S-ReLU surpasses GELU, underscoring its broad applicability across both vision and language domains and its potential to enhance stability and scalability.

## 1 INTRODUCTION

The expressive capacity and optimization dynamics of artificial neural networks are largely influenced by activation functions, which introduce nonlinearity to transform linear computations into complex representations and capture complex data patterns. Over the past three decades, more than 400 activation functions have been proposed to enhance network performance and efficiency (Kunc & Kléma, 2024). A clear evolutionary trend has emerged: a shift from nonsmooth activation functions toward more smooth ones. The key reason is that the non-differentiability of nonsmooth functions at certain points presents theoretical challenges for optimization algorithms and constitutes a practical bottleneck limiting performance improvements. In contrast, a growing body of theoretical work shows that smooth activation functions can deliver superior performance. Their smooth functional curves lead to flatter loss landscapes, which guide optimizers away from poor local minima, facilitate convergence to better solutions, and improve training stability. However, most existing smooth functions are exploratory modifications that address nonsmoothness but do not necessarily preserve the proven representational strengths of the original activation functions. Hence, developing a systematic framework for reshaping nonsmooth activation functions holds considerable value for machine learning research.

> **Major Questions**
>
> **1.** *Is there a systematic method to smooth non-smooth activation functions while retaining their original desirable properties?*
> **2.** *If yes, is there a relationship between the degree of smoothing and its gradient stability?*
> **3.** *Can an example demonstrate how the proposed methodology is applied in practice?*

Our work addresses the three research questions outlined above. We first define a smoothing kernel and apply it to transform a non-smooth activation into a smooth one. After verifying its smoothness and approximation, we analyze its Lipschitz constant and establish the connection between smoothness and gradient stability. To address the third question, we propose S-ReLU as a polished variant of the traditional ReLU. We evaluate its effectiveness on Vision Transformer (Dosovitskiy et al., 2020), its related derivatives (Han et al., 2021; Touvron et al., 2021), and ConvNeXt (Liu et al., 2022), across CIFAR-10, CIFAR-100 (Krizhevsky et al., 2009), and ImageNet-1K (Deng et al., 2009). The results consistently show that S-ReLU outperforms baseline activation functions. Moreover, fine-tuning experiments on large language models (LLMs) using Direct Preference Optimization (DPO) (Rafailov et al., 2023) further demonstrate that S-ReLU surpasses GELU, highlighting its broad applicability in practical scenarios. The principal contributions of this work are summarized below:

- **First, we show that the mollification can smoothly transform non-smooth activation functions while preserving the desirable properties of the original.** This ensures that the resulting activation not only inherits the advantages of the base function but also benefits from smoothness.

- **Second, we establish that higher degrees of smoothness lead to greater stability of training gradients, and we derive a quantitative relationship between the two.** This provides a theoretical foundation for subsequent studies on activation functions.

- **Third, we introduce S-ReLU, a new activation function derived from ReLU via mollification.** Extensive experiments across architectures, datasets, and tasks demonstrate that S-ReLU consistently outperforms existing activation functions and achieves state-of-the-art performance.

## 2 RELATED WORK

The study of activation functions has evolved through several stages. Early works introduced compression functions such as Sigmoid and Tanh (Hornik, 1991), which bound outputs to a finite interval but often saturate under extreme inputs, leading to gradient vanishing. To address these limitations, non-compression functions were proposed, with the rectified linear unit (ReLU) (Nair & Hinton, 2010) marking a landmark contribution. By preserving a unit gradient in the positive region and truncating negative activations to zero, ReLU greatly improved optimization stability. However, its asymmetric form results in structural drawbacks: neurons can become permanently inactive ("dying ReLU"), and the absence of negative outputs introduces distributional bias in subsequent layers.

To overcome these issues, a variety of variants were developed. Leaky ReLU (Maas et al., 2013) introduces a fixed negative slope to maintain gradient flow for sub-zero inputs, while Parameterized ReLU (PReLU) (He et al., 2015) adapts this slope as a learnable parameter. The Exponential Linear Unit (ELU) (Clevert et al., 2015) produces smooth negative outputs that reduce mean-shift effects, and the Continuously Differentiable Exponential Linear Unit (CELU) (Barron, 2017) simplifies the ELU parameterization for easier adjustment. More recently, smoother activations have gained traction for their theoretical and empirical benefits. Swish (SiLU) (Ramachandran et al., 2017) leverages a smooth and differentiable form to improve gradient propagation and training stability, while Mish (Misra, 2019) combines unbounded positive outputs with moderated negative values, facilitating deeper signal transmission and enhancing generalization.

On the applied side, GELU (Hendrycks & Gimpel, 2016) and its variants have become the dominant activations in large-scale vision and language architectures. They are the default in BERT and RoBERTa (Devlin et al., 2019), used in ViT (Dosovitskiy et al., 2020), and widely believed to support successive GPT models (Radford et al., 2019; Achiam et al., 2023) and recent models like PaLM (Chowdhery et al., 2023), LLaMA (Touvron et al., 2023), and DeepSeek-V2 (Liu et al., 2024). This reliance underscores the role of smooth activations in scaling and stabilizing deep learning. By contrast, a few open-source models, notably Mixtral(Jiang et al., 2024), adopt SiLU, reflecting continued exploration of alternatives.

## 3 MOTIVATION

Although piecewise linear activations such as ReLU are widely adopted for their simplicity and effectiveness, they suffer from intrinsic drawbacks that hinder further progress in deep neural networks. These include gradient instability arising from non-differentiability (Biswas et al., 2022; Lee,

2023), inefficient information propagation that accelerates signal degradation in deep layers (Hayou et al., 2019), and limited theoretical guarantees for regularization.

Our key idea is to build on the proven representational strengths of established activations and enhance them through systematic smoothing. By introducing smoothness while retaining the core advantages of the original function, we aim to improve differentiability, stability of gradient-based optimization, information propagation across layers, and regularization properties simultaneously. This perspective shifts the focus from designing entirely new exploratory functions to refining and elevating existing well-validated ones. Such an approach promises not only stronger and more stable performance in conventional deep learning tasks, but also provides a principled foundation for advancing modern large-scale architectures, including contemporary vision and language models.

## 4 METHODOLOGY

### 4.1 SMOOTH ACTIVATION FUCTION IS BETTER

Research by Hayou et al. (Hayou et al., 2019) indicates that the smoothness of an activation function is a key factor influencing the effective propagation of information in deep neural networks. A sufficient condition for an activation function to be smooth is that its second-order derivative can be piecewise represented as a sum of continuous functions. For smooth activation functions, the correlation between inter-layer neuron outputs converges to 1 at a rate of $O(1/l)$, with the specific formula being $1 - c^l \sim \beta_q/l$, where $c$ represents the correlation, $l$ is the number of network layers, and $\beta_q$ is a coefficient determined by the target variance $q$ and the activation function $f$. In contrast, for non-smooth functions like ReLU, this correlation converges at a rate of $O(1/l^2)$, with the specific relationship being $1 - c^l \sim 9\pi^2/2l^2$. This research reveals that in deep networks employing non-smooth activation functions, the correlation of neuron outputs rapidly approaches 1 as the network depth increases. This high degree of correlation impedes the effective propagation of information, leading to unstable gradients and diminished expressive power, ultimately impairing model performance. Therefore, selecting activation functions that meet specific smoothness requirements is an important strategy for enhancing both the training efficiency and the final performance of deep learning models.

### 4.2 SMOOTHED KERNEL FUNCTION

To smooth non-smooth activation functions, we can use the mollification method. This approach involves smoothing the target function by convolving it with a specific kernel. The goal is to create a smooth approximation of the original function without sacrificing its valuable attributes.

**Definition 1.** *We define smoothing kernel $\phi(x)$ as follows:*

$$\phi(x) = \begin{cases} Ae^{\frac{1}{x^2-1}}, |x| < 1 \\ 0, |x| \geq 1 \end{cases}$$

*Where constant A is defined as*

$$A = \left( \int_{-1}^{1} e^{\frac{1}{x^2-1}} dx \right)^{-1}$$

**Proposition 1.** *The smoothing kernel $\phi(x)$ is normalized, i.e., $\int_{\mathbb{R}} \phi(u)du = 1$.*

**Proposition 2.** *$\phi(x) \in C^\infty(\mathbb{R})$, i.e., $\phi(x)$ is infinitely differentiable on $\mathbb{R}$, and $\phi^{(k)}(x) = 0, |x| \geq 1$. Furthermore, for each natural number $k$, $\phi^{(k)}(x)$ is bounded on $\mathbb{R}$.*

Detailed proofs of proposition1 and proposition2 can be found in Appendix B.

**Definition 2.** *For any $\delta > 0$ we define a family of smoothing kernels $\{\phi_\delta\}_{\delta>0}$ generated by scaling the smoothing kernel $\phi(x)$ as follows:*

$$\phi_\delta(x) = \frac{1}{\delta} \phi\left(\frac{x}{\delta}\right)$$

This family of functions has several important properties. Each $\phi_\delta(x)$ is normalized, maintaining $\int_{\mathbb{R}} \phi_\delta(x)dx = 1$, and its support is scaled to the interval $[-\delta, \delta]$. As $\delta \to 0$, the family $\phi_\delta(x)$ converges to the Dirac delta function in the sense of distributions. We can now employ this framework to smooth the target activation function.

## 4.3 SMOOTHNESS OF THE MOLLIFIED ACTIVATION FUNCTION

**Answer to Question 1:** We show that the mollification can smoothly transform non-smooth activation functions while preserving the desirable properties of the original.

**Definition 3.** *Let $f(x)$ be a locally integrable activation function. The convolution of $f(x)$ with the smoothing kernel $\phi_\delta$ is called a smoothing of $f(x)$, denoted by $f_\delta(x)$ or $f * \phi_\delta$, and defined as:*

$$f_\delta(x) = (f * \phi_\delta)(x) = \int_{\mathbb{R}} f(y)\phi_\delta(x-y)dy = \int_{\mathbb{R}} f(x-y)\phi_\delta(y)dy$$

Let $H(x,y) = f(y)\phi_\delta(x-y)$. Taking partial derivative of it, we can get:

$$\frac{\partial^k}{\partial x^k}H(x,y) = \frac{\partial^k}{\partial x^k}[f(y)\phi_\delta(x-y)] = f(y)\frac{\partial^k}{\partial x^k}\phi_\delta(x-y) = f(y)\phi_\delta^{(k)}(x-y)$$

This partial derivative exists for any $k \geq 1$, because $\phi_\delta$ is a $C^\infty$ function according to Proposition 2. Let $x_0 \in \mathbb{R}$ be a fixed, arbitrary point. We will consider values of $x$ within a neighborhood of $x_0$. The function $\phi_\delta^{(k)}$ is continuous and has compact support, which implies that it is bounded on the $\mathbb{R}$. Let

$$M_k = \sup_{z \in \mathbb{R}}|\phi_\delta^{(k)}(z)| < \infty \tag{1}$$

The support of the integrand with respect to $y$ is the interval $[x-\delta, x+\delta]$. For any $x$ in the chosen neighborhood, this interval is contained in the larger compact set $K = [x_0 - 1 - \delta, x_0 + 1 + \delta]$. From Equation 1, we obtain the following bound:

$$\left|\frac{\partial^k}{\partial x^k}H(x,y)\right| = \left|f(y)\phi_\delta^{(k)}(x-y)\right| \leq |f(y)| \cdot M_k \cdot \chi_K(y)$$

where $\chi_K$ is the characteristic function of the set $K$, its specific form is as follows:

$$\chi_K(y) = \begin{cases} 1, & \text{if } y \in K \\ 0, & \text{if } y \notin K \end{cases}$$

Since $f \in L^1_{\text{loc}}(\mathbb{R})$, it is integrable on the compact set $K$. This implies that the dominating function $|f(y)| \cdot M_k \cdot \chi_K(y)$ is integrable. This argument shows that for any order k, the control conditions required for the differential operator $\frac{d^k}{dx^k}$ are satisfied. According to Lebesgue Dominated Convergence Theorem, we can move any order differential operator into the integral sign. Therefore, we can directly calculate the $k$-th derivative of $f_\delta$:

$$f_\delta^{(k)}(x) = \frac{d^k}{dx^k}f_\delta(x) = \frac{d^k}{dx^k}\int_{\mathbb{R}} f(y)\phi_\delta(x-y)dy = \int_{\mathbb{R}} \frac{\partial^k}{\partial x^k}[f(y)\phi_\delta(x-y)]dy$$

$$= \int_{\mathbb{R}} f(y)\phi_\delta^{(k)}(x-y)dy = (f * \phi_\delta^{(k)})(x)$$

This implies that the $k$-th derivative of $f_\delta$ exists for any positive integer $k$. Let

$$g_k(x) := f_\delta^{(k)}(x) = (f * \phi_\delta^{(k)})(x)$$

Since the function $\phi_\delta^{(k)}(x)$ is continuous with compact support, it is uniformly continuous on $\mathbb{R}$. Consider an arbitrary $h \in \mathbb{R}$, we then have

$$|g_k(x+h) - g_k(x)| = \left|\int_{\mathbb{R}} f(y)[\phi_\delta^{(k)}(x+h-y) - \phi_\delta^{(k)}(x-y)]dy\right|$$

$$\leq \int_{\mathbb{R}} |f(y)| \cdot |\phi_\delta^{(k)}(x+h-y) - \phi_\delta^{(k)}(x-y)|dy \tag{2}$$

Then we establish a dominating function for the integrand. By the triangle inequality,

$$|f(y)| \cdot |\phi_\delta^{(k)}(x+h-y) - \phi_\delta^{(k)}(x-y)| \leq |f(y)| \cdot (|\phi_\delta^{(k)}(x+h-y)| + |\phi_\delta^{(k)}(x-y)|) \leq |f(y)| \cdot 2M_k \cdot \chi_K(y)$$

Due to the uniform continuity of $\phi_\delta^{(k)}$, when $h \to 0$, we have

$$\lim_{h \to 0} |\phi_\delta^{(k)}(x+h-y) - \phi_\delta^{(k)}(x-y)| = 0 \tag{3}$$

Now we can apply Equation 3 and the Dominated Convergence Theorem to the Equality 2:

$$\lim_{h \to 0} |g_k(x+h) - g_k(x)| \leq \int_{\mathbb{R}} \lim_{h \to 0} \left( |f(y)| \cdot |\phi_\delta^{(k)}(x+h-y) - \phi_\delta^{(k)}(x-y)| \right) dy = 0$$

Thus, we have proven that for any $k \geq 0$, the derivative $f_\delta^{(k)}(x)$ is continuous, which implies $f_\delta(x) \in C^\infty(\mathbb{R})$. This establishes that the mollification process transforms the original activation function $f(x)$ into a smooth approximation $f_\delta(x)$.

### 4.4 Approximation Properties of the Mollified Activation Function

While achieving smoothness, a crucial question arises: to what degree does the new function $f_\delta(x)$ retain the core properties of the original function $f(x)$? Have we "distorted" the essence of the activation function in pursuit of smoothness? Let's discuss these questions next.

To analyze the approximation error, we start with the term $|f_\delta(x) - f(x)|$. Since the smoothing kernel integrates to 1, we can rewrite $f(x)$ in the following form:

$$f(x) = f(x) \cdot 1 = f(x) \int_{\mathbb{R}} \phi_\delta(u) du = \int_{\mathbb{R}} f(x) \phi_\delta(u) du \tag{4}$$

For the term $f_\delta(x)$, recall its definition and perform a substitution by letting $u = x - y$, which implies $y = x - u$. Thus,

$$f_\delta(x) = \int_{-\infty}^{\infty} f(y) \phi_\delta(x-y) dy = \int_{\infty}^{-\infty} f(x-u) \phi_\delta(u)(-du) = \int_{-\infty}^{\infty} f(x-u) \phi_\delta(u) du \tag{5}$$

Combining Equation 4 and Equation 5, we get:

$$|f_\delta(x) - f(x)| = \left| \int_{-\infty}^{\infty} f(x-u) \phi_\delta(u) du - \int_{-\infty}^{\infty} f(x) \phi_\delta(u) du \right| = \left| \int_{-\infty}^{\infty} [f(x-u) - f(x)] \phi_\delta(u) du \right|$$

According to the triangle inequality for integrals and the fact that $\phi_\delta(u) \geq 0$:

$$|f_\delta(x) - f(x)| \leq \int_{-\infty}^{\infty} |f(x-u) - f(x)| \phi_\delta(u) du = \int_{-\delta}^{\delta} |f(x-u) - f(x)| \phi_\delta(u) du$$

We consider the activation function $f : \mathbb{R} \to \mathbb{R}$ that is continuous. Let $D \subset \mathbb{R}$ be an arbitrary compact set. Since $u \in [-\delta, \delta]$, both $x$ and $x - u$ lie within the larger compact set $D' = D - [-\delta, \delta] = \{a - b \mid a \in D, b \in [-\delta, \delta]\}$. By the Heine-Cantor Theorem, a function that is continuous on a compact set is also uniformly continuous. Therefore, $f(x)$ is uniformly continuous on $D'$. By the definition of uniform continuity, for any $\varepsilon > 0$ given at the beginning, there must exist a $\Delta > 0$ such that whenever $|u| < \Delta$, $|f(z-u) - f(z)| < \varepsilon$ holds for all $z \in D'$. We choose our smoothing parameter $\delta$ to be less than $\Delta$ given by uniform continuity, i.e., $0 < \delta < \Delta$. In this way, for all $u \in [-\delta, \delta]$ in our integral expression, we have $|u| \leq \delta < \Delta$. Therefore, for these $u$, applying the property of uniform continuity yields the following:

$$|f(x-u) - f(x)| < \varepsilon \tag{6}$$

Substitute Equation 6 back into our integral:

$$|f_\delta(x) - f(x)| \leq \int_{-\delta}^{\delta} |f(x-u) - f(x)| \phi_\delta(u) du < \int_{-\delta}^{\delta} \varepsilon \cdot \phi_\delta(u) du = \varepsilon \int_{-\infty}^{\infty} \phi_\delta(u) du = \varepsilon$$

In summary, we have proven that for any $\varepsilon > 0$, there exists a $\Delta > 0$ such that whenever $0 < \delta < \Delta$, then $|f_\delta(x) - f(x)| < \varepsilon$ holds for all $x \in D$. From this, we can conclude the uniform convergence of the smoothed activation function $f_\delta(x)$ to the original activation function $f(x)$ on the set $D$.

By choosing a sufficiently small smoothing parameter $\delta$, we can make the smoothed activation function approximate the original function arbitrarily accurately. Uniform convergence plays a key role by ensuring that the smoothed activation function maintains the desirable characteristics of the original. It also ensures that the maximum error between the two functions approaches zero over any input interval of interest, thus avoiding unexpected deviations in specific regions.

## 4.5 LIPSCHITZ CONTINUITY ANALYSIS

> **Answer to Question 2:** We establish that higher degrees of smoothness lead to greater stability of training gradients, and we derive a quantitative relationship between the two.

In the field of deep learning, an abstract mathematical concept—Lipschitz continuity—is increasingly becoming a key factor in building more reliable, robust, and generalizable neural network models. From theoretical analysis to practical applications, Lipschitz continuity provides a powerful tool for deeply understanding and effectively controlling the behavior of deep networks. Lipschitz continuity offers a stronger condition than standard continuity by constraining a function's maximum rate of change. To understand this property, we will start with its formal definition.

**Definition 4.** *(Lipschitz Continuous Function)(Khromov & Singh, 2024)A function $f : \mathbb{R}^d \to \mathbb{R}^K$ with domain $dom(f) \subseteq \mathbb{R}^d$ is said to be C-Lipschitz continuous with respect to an $\alpha$-norm for some constant $C > 0$ if the following condition holds for all $x, y \in dom(f)$: $\|f(x) - f(y)\|_\alpha \le C\|x - y\|_\alpha$*

In our analysis, we concentrate on the smallest value of $C$ that satisfies the aforementioned condition. This value is formally defined as the Lipschitz constant. Lipschitz continuity of the activation function is crucial for ensuring well-behaved optimization, thereby promoting efficient convergence during training. Ensuring model stability by controlling the Lipschitz property is an effective way to prevent gradient runaway(Erichson et al.; Fazlyab et al., 2019; Gamba et al., 2023; Latorre et al.). The smaller the Lipschitz constant, the more stable the training gradients(Zhou et al., 2019; Khromov & Singh, 2024).

From the previous derivation, we know that the support set of the smoothing kernel function $\phi_\delta(x)$ is $[-\delta, \delta]$. The support set of a function derivative must be contained within the support set of the original function. Therefore, the support set of $\phi'_\delta(x)$ is also contained in $[-\delta, \delta]$.So we have

$$f'_\delta(x) = (f * \phi'_\delta)(x) = \int_{-\infty}^{\infty} f(y)\phi'_\delta(x-y)dy = \int_{x-\delta}^{x+\delta} f(y)\phi'_\delta(x-y)dy$$

For a continuous function $f(x)$, it must be bounded within the closed interval $[x-\delta, x+\delta]$, so we can find a constant $M > 0$ that satisfies the condition $|f(y)| \le M$. Substitute into the above equation and simplify:

$$|f'_\delta(x)| \le \int_{x-\delta}^{x+\delta} |f(y)||\phi'_\delta(x-y)|dy \le M \int_{x-\delta}^{x+\delta} |\phi'_\delta(x-y)|dy \tag{7}$$

Now let's calculate the integral $\int_{x-\delta}^{x+\delta} |\phi'_\delta(x-y)|dy$. We know that $\phi'_\delta(u) = \frac{1}{\delta^2}\phi'(\frac{u}{\delta})$, by variable substitution $v = (x-y)/\delta$:

$$\int_{x-\delta}^{x+\delta} |\phi'_\delta(x-y)|dy = \int_{x-\delta}^{x+\delta} \frac{1}{\delta^2}\left|\phi'\left(\frac{x-y}{\delta}\right)\right|dy = \frac{1}{\delta}\int_{-1}^{1} |\phi'(v)|dv \tag{8}$$

The value of this integral $\int_{-1}^{1} |\phi'(v)|dv$ is completely determined by the kernel function $\phi(x)$ we initially chose. It is a constant that is independent of $f$, $\delta$, and $x$, we denote this constant as $C_\phi$. Combining Equation 7 and Equation 8, we can conclude that

$$|f'_\delta(x)| \le M \cdot \left(\frac{1}{\delta}C_\phi\right) = \frac{M \cdot C_\phi}{\delta}$$

This shows that the Lipschitz constant of $f_\delta(x)$, $L_\delta$, satisfies: $L_\delta \le \frac{M \cdot C_\phi}{\delta}$.An increase in the value of $\delta$ leads to a broader refinement range for the original activation function, consequently enhancing its overall smoothness. Additionally, increasing the value of $\delta$ lowers the Lipschitz upper bound of the enhanced activation function, thereby promoting a more stable training gradient. Therefore, we can conclude: the higher the smoothness, the higher the training gradient stability, and there is a quantitative relationship between the two, where the specific quantitative relationship is determined by constants $M$ and $C_\phi$. This conclusion provides a principled guiding framework for the design of activation functions in the future.

## 4.6 RESHAPING RELU(S-RELU): FROM RELU TO BETTER

> **Answer to Question 3:** We introduce S-ReLU, a new activation function derived from ReLU via mollification.

In Section 4.1, we provide evidence that smoother activation functions are advantageous for enhancing both the efficiency of the training process and the final performance of the model. Sections 4.3 and 4.4 further establish that activation functions constructed via smoothing theory possess desirable smoothness and approximation properties; specifically, they are infinitely differentiable and can approximate the original activation function arbitrarily closely, thereby preserving its favorable characteristics. Building upon these findings, we next apply the proposed methodology to a concrete instance. In particular, we select the widely used ReLU function as the basis for mollification. Regarding the choice of kernel, we strike a balance between theoretical rigor and engineering practicality by adopting the Epanechnikov kernel, which is theoretically optimal for minimizing the Mean Integrated Squared Error. Moreover, among the class of non-negative polynomials satisfying the fundamental smoothness conditions, it exhibits the lowest degree and the simplest closed form.

The standard ReLU activation function computes the maximum of zero and its input, as given by $f(x) = \text{ReLU}(x) = \max(0, x)$. We select the Epanechnikov kernel function, denoted as $\phi_\delta(x)$, parameterized by the smoothing radius $\delta$. This kernel acts as a weighting function with compact support on the interval $[-\delta, \delta]$. Its normalized form is:

$$\phi_\delta(x) = \begin{cases} \frac{3}{4\delta}\left(1 - \frac{x^2}{\delta^2}\right) & \text{,if } |x| \le \delta \\ 0 & \text{,if } |x| > \delta \end{cases}$$

Applying the preceding theory yields the smoothed activation function S-ReLU:

$$f_\delta(x) = (f * \phi_\delta)(x) = \int_R \max(0, y)\phi_\delta(x - y)dy = \begin{cases} 0 & \text{,if } x \le -\delta \\ \frac{x}{2} + \frac{3x^2}{8\delta} + \frac{3\delta}{16} - \frac{x^4}{16\delta^3} & \text{,if } -\delta < x < \delta \\ x & \text{,if } x \ge \delta \end{cases}$$

Differentiating the S-ReLU function twice yields the following:

$$f_\delta''(x) = \begin{cases} \frac{3(\delta^2 - x^2)}{4\delta^3} & \text{, if } -\delta < x < \delta \\ 0 & \text{, if } |x| \ge \delta \end{cases}$$

This indicates that S-ReLU is an activation function with continuous second derivatives, satisfying the definition of a smooth function in Section 4.1. We use the uniform error metric to quantify how well S-ReLU approximates the target function. The calculation result is given as follows:

$$||\text{S-ReLU}(x) - \text{ReLU}(x)||_\infty = \sup_{x \in \mathbb{R}} |\text{S-ReLU}(x) - \text{ReLU}(x)| = \frac{3\delta}{16}$$

This result clearly demonstrates that, compared to ReLU, the approximation error of S-ReLU is controllable and proportional to the smoothing radius $\delta$. This is a valuable property, as it allows us to precisely control the extent of the approximation error introduced by the smoothing operation by choosing the value of $\delta$. A detailed proof is provided in the Appendix C. Further details regarding S-ReLU are available in the appendices. Specifically, Appendix D presents the Python-style pseudocode, while Appendix E contains a more in-depth discussion of its characteristics. Finally, we discuss Lipschitz continuity. Based on the theory in section4.5, we can calculate the Lipschitz constant for each activation function.

**Fact 1.** *Lipschitz constant of GELU is 1.084; Lipschitz constant of SiLU is 1.100; Lipschitz constant of Mish is 1.089. Lipschitz constant of S-ReLU is 1.000.*

The detailed proof can be found in Appendix F. S-ReLU's Lipschitz constant of 1 ensures that its output never changes more rapidly than its input. This property acts as a vital safeguard for stabilizing gradient flow in the network, which helps prevent the problem of exploding gradients and makes the model more robust to small input perturbations—a clear advantage over other activation functions.

## 5 EXPERIMENTS

**Experimental Setup.** We conducted experiments on the CIFAR-10, CIFAR-100, and ImageNet-1K datasets to evaluate the effectiveness of S-ReLU for image classification, as well as on LLM fine-tuning with human preference datasets SHP (Ethayarajh et al., 2022), HH (Bai et al., 2022), and GPT-2(Radford et al., 2019). We compare against representative ReLU variants, including GELU, ELU, PReLU, CELU, SiLU, and Mish. Detailed training configurations and hyperparameters are provided in Appendix H.

### 5.1 TASK OF IMAGE CLASSIFICATION

**Evaluation of ViTs on CIFAR-10, CIFAR-100, and ImageNet-1K.** To comprehensively evaluate the proposed activation, we conducted experiments on ViT , DeiT and TNT. CIFAR-10 and CIFAR-100 were selected to examine the sensitivity of activation functions under different data distributions, while ImageNet-1K, with its larger image resolution and broader category coverage, was employed to assess performance in more challenging large-scale scenarios.

As summarized in Table 1, S-ReLU consistently outperforms all existing ReLU variants across every dataset and architecture. The gains are evident on both small-scale benchmarks(CIFAR-10/100), where S-ReLU achieves markedly higher accuracy, and on the large-scale ImageNet-1K dataset, where it surpasses strong baselines under more demanding conditions. These results demonstrate not only the superior fitting ability of S-ReLU, but also its strong generalization capacity, showing that the advantages of smoothing are preserved across data regimes of different scales and complexities.

Table 1: Test accuracy on CIFAR-10, CIFAR-100, and ImageNet-1K over 100 epochs.

| Top-one Accuracy | | GELU | ELU | PReLU | CELU | SiLU | Mish | S-ReLU |
|---|---|---|---|---|---|---|---|---|
| CIFAR-10 | ViT-Tiny | 70.4±0.2 | 66.4±0.5 | 78.0±0.6 | 66.5±0.6 | 68.6±0.3 | 68.7±0.3 | **81.0±0.6** |
| CIFAR-10 | DeiT-Tiny | 72.4±0.7 | 67.6±0.6 | 75.4±0.1 | 67.7±0.8 | 69.9±0.5 | 70.2±0.6 | **81.1±0.3** |
| CIFAR-10 | TNT-Small | 73.7±0.5 | 69.5±0.6 | 75.8±0.3 | 68.7±0.2 | 71.1±0.7 | 71.6±0.8 | **84.8±0.2** |
| CIFAR-10 | Average | 72.2±0.5 | 67.8±0.6 | 76.4±0.3 | 67.6±0.5 | 69.9±0.5 | 70.2±0.6 | **82.3±0.4** |
| CIFAR-100 | ViT-Tiny | 32.6±0.8 | 28.9±0.1 | 43.2±1.0 | 28.9±0.2 | 31.2±0.6 | 30.6±0.8 | **51.2±0.6** |
| CIFAR-100 | DeiT-Tiny | 46.6±0.9 | 56.9±0.0 | 50.0±0.5 | 40.5±0.5 | 43.5±0.6 | 43.8±1.0 | **57.1±0.2** |
| CIFAR-100 | TNT-Small | 47.5±0.8 | 43.6±0.3 | 49.0±0.7 | 43.0±0.5 | 45.0±0.9 | 45.5±0.8 | **61.6±0.4** |
| CIFAR-100 | Average | 42.2±0.8 | 43.1±0.1 | 47.4±0.7 | 37.5±0.4 | 39.9±0.7 | 40.0±0.9 | **56.6±0.3** |
| ImageNet-1K | ViT-Tiny | 53.9±0.3 | 37.2±0.6 | 56.8±0.3 | 37.6±0.5 | 46.1±0.7 | 46.9±1.1 | **56.6±0.4** |
| ImageNet-1K | DeiT-Tiny | 61.7±0.4 | 49.1±0.7 | 60.8±0.4 | 48.9±0.8 | 58.5±0.7 | 58.9±0.3 | **64.9±0.4** |
| ImageNet-1K | Average | 57.8±0.4 | 43.2±0.7 | 58.8±0.4 | 43.3±0.7 | 52.3±0.7 | 52.9±0.7 | **60.8±0.4** |

**Evaluation of ConvNeXt on CIFAR-10, CIFAR-100 and ImageNet-1K.** Beyond transformer families, we investigate the universality of S-ReLU in convolutional networks by testing it on Con-vNeXt, a leading convolutional model designed with modern principles to rival Transformers. This setting provides a stringent test of whether the performance improvements of S-ReLU are tied to specific architectural choices or generalize broadly across models.

Experimental results on CIFAR-10 and CIFAR-100 with 100 training epochs demonstrate that Con-vNeXt models equipped with S-ReLU achieve consistently higher classification accuracy than those with existing activation functions. Notably, on the more challenging ImageNet-1K benchmark, S-ReLU continues to surpass GELU, SiLU, Mish, and other baselines, establishing new performance levels for ConvNeXt. These results confirm that the advantages of S-ReLU are not restricted to Transformer-based architectures but extend robustly to convolutional networks as well. Together with our earlier findings on Vision Transformers, these results highlight that S-ReLU delivers both architectural universality and strong generalization ability.

Table 2: Test accuracy of experiments conducted on ConvNeXt-tiny for 100 epochs.

| Top-one Accuracy | | GELU | ELU | PReLU | CELU | SiLU | Mish | S-ReLU |
|---|---|---|---|---|---|---|---|---|
| CIFAR-10 | ConvNeXt | 64.9±0.4 | 59.8±0.5 | 64.6±1.4 | 59.8±0.5 | 60.6±0.2 | 61.4±0.4 | **89.9±0.1** |
| CIFAR-100 | ConvNeXt | 36.6±0.3 | 30.3±0.4 | 35.2±0.5 | 30.5±0.2 | 35.0±0.9 | 35.3±0.7 | **66.8±0.1** |
| ImageNet-1K | ConvNeXt | 72.9±0.3 | 71.7±0.5 | 72.9±0.5 | 71.8±0.9 | 72.3±0.7 | 72.8±0.6 | **73.1±0.2** |

## 5.2 Task of Large Language Model (LLM) Fine-tuning

To test the generalizability of our proposed S-ReLU activation function outside of computer vision, we also evaluated its performance in the increasingly important field of Large Language Models. Specifically, we fine-tune GPT-2 on the SHP and HH datasets using DPO. Importantly, both datasets pose challenges of stability and nuanced representation, making them suitable testbeds for evaluating whether activation functions like S-ReLU can enhance optimization robustness and expressive capacity. Because DPO relies on a reference strategy that may diverge from the true data distribution, we begin with supervised fine-tuning (SFT) to reduce this gap and then apply DPO with different penalty coefficients $\beta \in 0.1, 2, 5$.

Table 3 reports the mean and standard deviation for each evaluation metric, averaged over several experimental runs. Compared with GELU, S-ReLU consistently achieves higher chosen rewards, lower rejected rewards, larger reward margins, and improved preference accuracy. These results indicate that S-ReLU not only enhances the model's ability to assign higher utility to preferred responses while penalizing non-preferred ones, but also strengthens its discriminative margin and alignment with human feedback. The consistent performance gains on all metrics confirm that introducing smoothness is an effective strategy, especially since it maintains the representational capabilities of the initial activation function. This demonstrates that the advantages of S-ReLU generalize robustly from vision tasks to LLM preference optimization, thereby confirming its potential as a broadly applicable activation function for deep learning.

Table 3: Metrics comparison between S-RELU and GELU in the task of LLM fine-tuning.

| Evaluation Metrics | | Chosen Reward | Rejected Reward | Margin Reward↑ | Preference Accuracy↑ |
|---|---|---|---|---|---|
| $\beta = 0.1$ | S-RELU | 0.1501±0.0005 | 0.0517±0.0003 | **0.0984±0.0008** | **0.5938±0.0000** |
| | GELU | -0.2037±0.0012 | -0.2696±0.0007 | 0.0659±0.0019 | 0.5313±0.0012 |
| $\beta = 2$ | S-RELU | 0.3861±0.0018 | 0.3329±0.0012 | **0.0532±0.0030** | **0.5156±0.0005** |
| | GELU | -1.6600±0.0003 | -1.7060±0.0052 | 0.0460±0.0055 | 0.5080±0.0022 |
| $\beta = 5$ | S-RELU | 1.1062±0.0025 | -0.7602±0.0016 | **1.8664±0.0041** | **0.5103±0.0017** |
| | GELU | -3.6484±0.0031 | -4.8299±0.0003 | 1.1815±0.0034 | 0.5012±0.0001 |

## 6 Discussion

Developing effective activation functions has long been a central problem in machine learning. While non-smooth activations suffer from well-documented limitations, smooth activations have emerged as a promising direction. Yet, a general methodology for systematically smoothing non-smooth activations remains absent. In this work, we introduce a mathematically rigorous and practically effective framework based on mollification theory to smooth non-smooth activations while preserving their desirable properties. Within this framework, we establish a quantitative relationship between smoothness and gradient stability, offering a theoretical foundation for advancing activation function design. Building on this, we derive a new activation function, S-ReLU, as a polished variant of ReLU. Across image classification and LLM fine-tuning tasks, S-ReLU consistently outperforms existing rectified ReLU variants. Our findings position S-ReLU as a strong new member of the family of high-performance activation functions, while also opening avenues for future research on smooth and stable architectures.

**Limitations and Future Work.** The results point to both theoretical and practical opportunities for advancing activation research. First, we employed the Epanechnikov kernel as the smoothing kernel due to its strong theoretical characterization, but alternative choices may yield improvements. In particular, shifting kernel design from fixed theoretical selection to dynamic, data-driven, or learnable formulations could provide greater flexibility and performance. Second, the applicability of mollification to other architectures, such as Kolmogorov–Arnold Networks (KANs), remains an open question. Third, although this work establishes the link between smoothness and gradient stability and analyzes the Lipschitz constant, deeper theoretical investigations are needed to understand the impact of smoothing on representational capacity, generalization bounds, and convergence. Finally, while S-ReLU demonstrates strong empirical results in image classification and large-scale language model fine-tuning, its extension to broader domains holds promise for advancing future studies.

# 7 ETHICS STATEMENT

All authors have read and agree to adhere to the ICLR Code of Ethics. We confirm that this work complies with the ethical standards outlined therein, and we explicitly acknowledge our commitment to these principles during the submission and review process.

# 8 REPRODUCIBILITY STATEMENT

All datasets used in this work are publicly available. Detailed descriptions of preprocessing, hyper-parameters, and training configurations are provided in Section 5 and Appendix H. An anonymous implementation of S-ReLU and experimental code is included in the supplementary material to facilitate full reproducibility.

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

## A USE OF LLMS

We used large language models (LLMs) solely for minor writing polish.

## B POOF OF PROPOSITION 1 AND 2

**Proposition 1.** The Smoothing Kernel has normalization:

$$\int_{\mathbb{R}} \phi(u)du = 1$$

*Proof.* The Smoothing Kernel has normalization

$$\int_{\mathbb{R}} \phi(u)du = \int_{-1}^{1} \phi(u)du = \int_{-1}^{1} Ae^{\frac{1}{u^2-1}}du = \frac{1}{\int_{-1}^{1} e^{\frac{1}{u^2-1}}du} \cdot \int_{-1}^{1} e^{\frac{1}{u^2-1}}du = 1$$

$\square$

**Proposition 2.** $\phi(x) \in C^\infty(\mathbb{R})$, i.e., $\phi(x)$ has continuous derivatives of any order, and $\phi^{(k)}(x) = 0, |x| \geq 1$, Furthermore, for each natural number $k, \phi^{(k)}(x)$ is bounded on $\mathbb{R}$.

*Proof.* $(a)$ For $|x| > 1$, In this interval, $\phi(x) = 0$. Any derivative of the constant function is 0, so $\phi(x)$ is $C^\infty$ in this interval.

$(b)$ For $|x| < 1, \phi(x)$ is an elementary function and naturally has derivatives of any order.

$(c)$ For $|x| = 1$, since $\phi(x)$ is an even function, we only need to consider case where $x = 1$. According to the definition, for $x \geq 1, \phi(x) = 0$, so all right-sided derivatives $\phi_+^{(k)}(1)$ are 0. Our task is to prove that all left-sided derivatives $\phi_-^{(k)}(1)$ are also 0. We use mathematical induction to prove that the proposition $P(k): \phi^{(k)}(1) = 0$ holds for all $k \geq 0$.

For $k = 0$, obviously valid. Assume that $P(k)$ holds for some $k \geq 0, i.e., \phi^{(k)}(1) = 0$. For $|x| < 1, \phi^{(k)}(x)$ takes the form: $\phi^{(k)}(x) = R_k(x)e^{\frac{1}{x^2-1}}$, where $R_k(x)$ is a polynomial with $x$ and $(x^2-1)^{-1}$ as variables, i.e., a rational function. We need to prove that $P(k+1)$ holds, i.e., $\phi^{(k)}(1) = 0$. According to the definition of derivatives

$$\phi^{(k+1)}(1) = \lim_{h \to 0} \frac{\phi^{(k)}(1+h) - \phi^{(k)}(1)}{h}$$

Since $\phi^{(k)}(1) = 0$ and for $h > 0, \phi^{(k)}(1+h) = 0$, the right limit is obviously 0. We only need to calculate the left limit:

$$\phi_-^{(k+1)}(1) = \lim_{h \to 0^-} \frac{\phi^{(k)}(1+h)}{h} = \lim_{x \to 1^-} \frac{\phi^{(k)}(x)}{x-1} = \lim_{x \to 1^-} \frac{R_k(x)}{x-1} e^{\frac{1}{x^2-1}} = 0$$

Therefore $\phi^{(k+1)}(1) = 0, P(k+1)$ holds true. According to mathematical induction, for all $k \geq 0, \phi^{(k)}(x) = 0$. Similarly, this also holds at $x = -1$. In summary, $(a), (b)$, and $(c)$ show that $\phi(x)$ is infinitely differentiable over the entire $\mathbb{R}$.

**Lemma B.1.** *(Extreme Value Theorem)A continuous function defined on a compact set must be bounded and can reach its maximum and minimum values.*

From the lemma, we know that there exists a constant $M_k > 0$ such that for all $x \in [-1, 1]$, we have $|\phi^{(k)}(x)| \leq M_k$. And for all $|x| \geq 1, \phi^{(k)}(x) = 0$. Therefore, for any $x \in \mathbb{R}$, we have $|\phi^{(k)}(x)| \leq M_k$. $\square$

## C  PROOFS RELATED TO S-RELU

First, prove the expression for S-ReLU.

$$f_\delta(x) = (f * \phi_\delta)(x) = \int_R \max(0, y)\phi_\delta(x - y)dy = \int_{\max(x-\delta,0)}^{x+\delta} y \cdot \phi_\delta(x-y)dy$$

(1)When $x \leq -\delta$, at this point, the right boundary of the integration window satisfies $x + \delta \leq 0$. This implies that the entire integration interval $[x - \delta, x + \delta]$ falls within the range $y \leq 0$. Within this interval, $f(y) = \max(0, y) = 0$. Therefore, the integral is:

$$f_\delta(x) = \int_{x-\delta}^{x+\delta} 0 \cdot \phi_\delta(x - y)dy = 0$$

(2)When $x \geq \delta$, at this point, the left boundary of the integration window satisfies $x - \delta \geqslant 0$. This means the entire integration interval $[x - \delta, x + \delta]$ lies within the range where $y > 0$. Within this interval, $f(y) = \max(0, y) = y$. The integral is:

$$\begin{aligned}
f_\delta(x) &= \int_{x-\delta}^{x+\delta} y \cdot \phi_\delta(x - y)dy \\
&= \int_\delta^{-\delta} (x - u)\phi_\delta(u)(-du) = \int_{-\delta}^{\delta} (x - u)\phi_\delta(u)du \\
&= x\int_{-\delta}^{\delta} \phi_\delta(u)du - \int_{-\delta}^{\delta} u\phi_\delta(u)du = x
\end{aligned}$$

(3)When $-\delta < x < \delta$, at this point, the integration interval $[x - \delta, x + \delta]$ spans the origin. According to our previous analysis, the lower limit of integration is $max(x - \delta, 0) = 0$, and the upper limit is $x + \delta$. The integral is:

$$\begin{aligned}
f_\delta(x) &= \int_0^{x+\delta} y \cdot \phi_\delta(x - y)dy \\
&= \int_x^{-\delta} (x - u)\phi_\delta(u)(-du) = \int_{-\delta}^{x} (x - u)\phi_\delta(u)du \\
&= \frac{3}{4\delta} \int_{-\delta}^{x} (x - u)\left(1 - \frac{u^2}{\delta^2}\right) du \\
&= \frac{x}{2} + \frac{3x^2}{8\delta} + \frac{3\delta}{16} - \frac{x^4}{16\delta^3}
\end{aligned}$$

Therefore, the expression for S-ReLU is

$$f_\delta(x) = \begin{cases}
0 & \text{,if } x \leq -\delta \\
\frac{x}{2} + \frac{3x^2}{8\delta} + \frac{3\delta}{16} - \frac{x^4}{16\delta^3} & \text{,if } -\delta < x < \delta \\
x & \text{,if } x \geq \delta
\end{cases}$$

Next, we will calculate the uniform error. When $x \leq -\delta$ or $x \geq \delta$: In these two intervals, the definition of $f_\delta(x)$ is identical to that of ReLU$(x)$, so the error is 0. When $0 \leq x < \delta$, the error function is given by

$$E(x) = |f_\delta(x) - x| = |\left(\frac{x}{2} + \frac{3x^2}{8\delta} + \frac{3\delta}{16} - \frac{x^4}{16\delta^3}\right) - x|$$

This function is monotonically decreasing on $[0, \delta]$, so its minimum value is $E(0) = \frac{3a}{16}$. Similarly, the same result holds for the interval $[-\delta, 0]$, so the uniform error is $\frac{3\delta}{16}$.

## D  SMOOTHED RELU(S-RELU) PSEUDOCODE

**Algorithm 1:** Smoothed ReLU(S-ReLU) Pseudocode

```
import torch
import torch.nn as nn
import torch.nn.functional as F

class SReLU(nn.Module):
  def __init__(self, trainable=False):
      super().__init__()
      super(SReLU, self).__init__()
  def forward(self, x):
      a = 0.01
      condition1 = (x <= -a)
      condition2 = (x > -a) & (x < a)
      condition3 = (x >= a)
      p1 = torch.zeros_like(x)
      p2 = (x/2.0 + 3.0*x**2/(8.0*a) + 3.0*a/16.0 - x**4/(16.0*a**3))
      p3 = x
      output = torch.where(condition1, p1,torch.where(condition2, p2,p3))
      return output
```

## E    FURTHER DICUSSION ON PROPERTIES OF S-RELU

Figure 1 shows S-ReLU images obtained for different values of $\delta$. When $\delta$ approaches 0, the function becomes the ReLU. As shown in the figure, the S-ReLU is smooth and differentiable everywhere, completely eliminating the non-differentiability of the ReLU at the origin, which is crucial for gradient-based optimization algorithms. Smooth activation functions produce smoother loss landscapes, theoretically helping to accelerate the optimization process and find better solutions. Within the interval $(-\delta, 0)$, the S-ReLU function value varies, while its gradient persists and is non-zero. This directly addresses the drawback of the standard ReLU, where the gradient is always zero on the negative half-axis, effectively avoiding the "dead ReLU problem"—a phenomenon in which neurons permanently stop learning due to improper weight updates $\delta$, as an adjustable hyperparameter, allows precise control of the degree of smoothing based on specific task requirements.

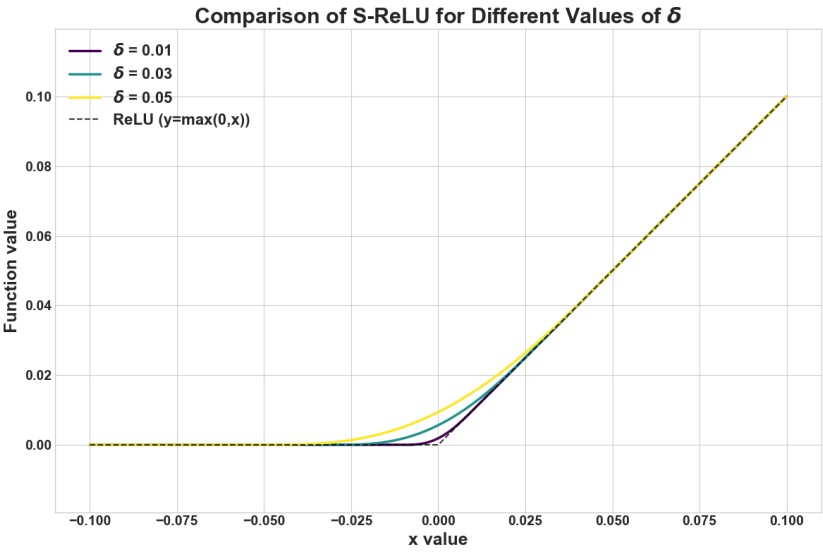

Figure 1: S-ReLU with different $\delta$ value

## F  PROOF OF LIPSCHITZ CONTINUITY ANALYSIS

**Remark 1.** *Lipschitz constant of GELU is 1.084.*

*Proof.* To establish the Lipschitz continuity of the GELU activation function, we first analyze its derivative to find its upper bound. The first derivative of GELU$(x)$ is defined as:

$$\frac{d\text{GELU}(x)}{dx} = \Phi(x) + x\phi(x) = \Phi(x) + x\frac{1}{\sqrt{2\pi}}e^{-\frac{x^2}{2}}$$

where $\Phi(x)$ is the cumulative distribution function and $\phi(x)$ is the probability density function of the standard normal distribution. To find the maximum value of this derivative, we utilize the second derivative test. The second derivative is calculated as:

$$\frac{d^2\text{GELU}(x)}{dx^2} = \phi(x)(2 - x^2) = \frac{1}{\sqrt{2\pi}}e^{-\frac{x^2}{2}}(2 - x^2)$$

The extrema of the first derivative occur where the second derivative equals zero. Setting $\frac{d^2\text{GELU}(x)}{dx^2} = 0$, we solve for $x$:

$$\frac{1}{\sqrt{2\pi}}e^{-\frac{x^2}{2}}(2 - x^2) = 0$$

Since the exponential term is always positive, the expression is zero only when $2 - x^2 = 0$. This yields the critical points $x = \pm\sqrt{2}$. By evaluating the first derivative at these critical points, we can determine its maximum value. For $x = \sqrt{2}$, we find:

$$\frac{d\text{GELU}(x)}{dx}\bigg|_{x=\sqrt{2}} = \Phi(\sqrt{2}) + \sqrt{2} \cdot \frac{1}{\sqrt{2\pi}}e^{-1} \approx 1.084$$

This value represents the maximum slope of the GELU function. Since the derivative is bounded by this value, we confirm that GELU is Lipschitz continuous with a Lipschitz constant of approximately 1.084. $\qquad\square$

**Remark 2.** *Lipschitz constant of SiLU is 1.100.*

*Proof.* To establish the Lipschitz continuity of the SiLU (Sigmoid Linear Unit) function, we first analyze its derivative to find its upper bound. The SiLU function is defined as:

$$\text{SiLU}(x) = x\sigma(x) = \frac{x}{1 + e^{-x}}$$

The Lipschitz constant is the maximum absolute value of its derivative, $\frac{d\text{SiLU}(x)}{dx}$. To find this maximum, we employ the second derivative test. The first and second derivatives are:

$$\frac{d\text{SiLU}(x)}{dx} = \frac{(x+1)e^{-x} + 1}{(1 + e^{-x})^2}$$

$$\frac{d^2\text{SiLU}(x)}{dx^2} = \frac{e^x(-e^x(x-2) + x + 2)}{(1 + e^x)^3}$$

The extrema of the first derivative occur where the second derivative equals zero. Setting $\frac{d^2\text{SiLU}(x)}{dx^2} = 0$, we solve for the critical points, which are found to be $x \approx \pm 2.3994$. By evaluating the first derivative at these critical points and analyzing its behavior, we find that its value is bounded within the interval $[-0.100, 1.100]$. The Lipschitz constant is the supremum of the derivative's absolute value. Therefore, we conclude that the Lipschitz constant for SiLU is approximately 1.100. $\qquad\square$

**Remark 3.** *Lipschitz constant of Mish is 1.089.*

*Proof.* To determine the Lipschitz constant for the Mish activation function, we perform an analysis of its derivatives. The objective is to find the supremum of the absolute value of the first derivative, which requires locating its global extrema. The Mish function is defined by the expression:

$$\text{Mish}(x) = x\frac{e^{2x} + 2e^x}{e^{2x} + 2e^x + 2}$$

The extrema of its first derivative are found by identifying the roots of the second derivative. The first and second derivatives are as follows:

$$\frac{d\text{Mish}(x)}{dx} = \frac{e^x[4(x+1) + 4e^{2x} + e^{3x} + e^x(4x+6)]}{(e^{2x} + 2e^x + 2)^2}$$

$$\frac{d^2\text{Mish}(x)}{dx^2} = \frac{4e^x(3e^{2x}(x-2) + 2e^{3x}(x-1) - 2(x+2) - 2e^x(x+4))}{(e^{2x} + 2e^x + 2)^3}$$

By setting the second derivative to zero, we solve for the critical points of the first derivative. This calculation yields two approximate solutions: $x_1 \approx -2.2564$ and $x_2 \approx 1.4906$. Evaluating the first derivative at these critical points and analyzing its global behavior reveals that its values are bounded within the interval $[-0.113, 1.089]$. Therefore, the Lipschitz constant for the Mish function, which is the maximum absolute value of its derivative, is established to be 1.089. $\qquad\square$

**Remark 4.** *Lipschitz constant of S-ReLU is 1.000.*

*Proof.* We know that the expression for SiLU:

$$\text{S}-\text{ReLU}(x) = \begin{cases} 0 & , \text{if} x \leq -\delta \\ \frac{x}{2} + \frac{3x^2}{8\delta} + \frac{3\delta}{16} - \frac{x^4}{16\delta^3} & , \text{if} -\delta < x < \delta \\ x & , \text{if} x \geq \delta \end{cases}$$

Calculating the first derivative yields:

$$\frac{d(\text{S}-\text{ReLU}(x))}{dx} = \begin{cases} 0 & , \text{if} x \leq -\delta \\ \frac{1}{2} + \frac{3x}{4\delta} - \frac{x^3}{4\delta^3} & , \text{if} -\delta < x < \delta \\ 1 & , \text{if} x \geq \delta \end{cases}$$

Calculating the second derivative yields:

$$\frac{d^2(\text{S}-\text{ReLU}(x))}{dx^2} = \begin{cases} \frac{3}{4\delta} - \frac{3x^2}{4\delta^3} & , \text{if} -\delta \leq x \leq \delta \\ 0 & , \text{otherwise} \end{cases}$$

We already know that on the interval $(-\infty, -\delta)$, $|\frac{d(\text{S}-\text{ReLU}(x))}{dx}| = 0$, and on the interval $(\delta, \infty)$, $|\frac{d(\text{S}-\text{ReLU}(x))}{dx}| = 1$. Now we need to analyze the extrema of the derivative on the interval $(-\delta, \delta)$. Let $g(x) = \frac{1}{2} + \frac{3x}{4\delta} - \frac{x^3}{4\delta^3}$. We find its critical points by taking the derivative of $g(x)$:

$$g'(x) = \frac{d}{dx}\left(\frac{1}{2} + \frac{3x}{4\delta} - \frac{x^3}{4\delta^3}\right) = \frac{3}{4\delta} - \frac{3x^2}{4\delta^3}$$

Setting $g'(x) = 0$ to find the fixed point yields $x = \pm\delta$. The fixed point lies on the boundary of the interval. This indicates that within the interval $(-\delta, \delta)$, $g(x)$ is monotonic. For any $x \in (-\delta, \delta)$, we have $x^2 < \delta^2$. Therefore, $g'(x) = \frac{3}{4\delta^3}(\delta^2 - x^2) > 0$. Thus, calculating the derivative at the endpoint yields:

$$\lim_{x \to -\delta^+} \frac{d(\text{S}-\text{ReLU}(x))}{dx} = \frac{1}{2} + \frac{3(-\delta)}{4\delta} - \frac{(-\delta)^3}{4\delta^3} = 0$$

$$\lim_{x \to \delta^-} \frac{d(\text{S}-\text{ReLU}(x))}{dx}(x) = \frac{1}{2} + \frac{3\delta}{4\delta} - \frac{\delta^3}{4\delta^3} = 1$$

Therefore, the derivative of S-ReLU is bounded within the interval $[0, 1]$. Therefore, S-ReLU's Lipschitz constant is 1.000. $\qquad\square$

## G    SENSITIVITY ANALYSIS OF PARAMETER $\delta$

In this section, we focus on the impact of different $\delta$ on the final results. We set $\delta$ to 0.001, 0.01, 0.1, 0.2, 0.5, 1, 5 and 10, and conduct experiments on the CIFAR10 and CIFAR100 datasets using ViT-tiny. We perform three runs and report the mean and standard deviation in Table 4.

Table 4: Test accuracy of 100 experiments using ViT at different sensitivity parameters

| $\delta$ | 0.001 | 0.01 | 0.1 | 0.2 | 0.5 | 1 | 5 | 10 |
|---|---|---|---|---|---|---|---|---|
| CIFAR-10 | 81.0±0.6 | 80.1±0.2 | 80.4±0.3 | 78.2±0.2 | 80.4±0.2 | 73.9±0.3 | 63.4±0.5 | 58.5±0.3 |
| CIFAR-100 | 51.2±0.6 | 48.7±0.3 | 48.7±0.5 | 48.5±0.5 | 45.5±0.3 | 41.4±0.6 | 43.8±1.0 | 25.7±0.2 |

# H DETAILS OF EXPERIMENTAL SETTINGS

In this appendix, we provide implementation details and hyperparameter settings to facilitate reproducibility. The main datasets and baseline methods are introduced in the main text (Section 5). Here we report additional training configurations specific to each model and dataset. Hyperparameter sensitivity to $\delta$ is reported in Appendix G; according to the experimental results, we use $\delta = 0.001$ for all experiments.

**Image classification.** For CIFAR-10 and CIFAR-100, we trained ViT-Tiny, DeiT-Tiny, and TNT-Small for 100 epochs. All models used AdamW with weight decay of 0.05, cosine annealing learning rate scheduling (initial learning rate $2.5 \times 10^{-4}$, minimum $1 \times 10^{-5}$, with 20 warmup epochs starting from $1 \times 10^{-6}$), and gradient clipping of 1.0. Training was performed with a batch size of 256, cross-entropy loss, and layer normalization, without dropout or drop path. For CIFAR tasks, images of size $32 \times 32$ were divided into patches of size 4, with embedding dimensions of 192 for ViT-Tiny and DeiT-Tiny and 384 for TNT-Small. Standard data augmentations provided by `timm` were applied.

For ImageNet-1K, we trained ViT-Tiny and DeiT-Tiny for 100 epochs under largely similar optimization settings. Images were resized to $224 \times 224$, patch size was set to 16, and the embedding dimension was 192. The same AdamW optimizer, learning rate schedule, batch size (256), loss function, and normalization strategy were adopted, with no dropout or drop path used.

**LLM fine-tuning.** For SHP, HH, and GPT-2 fine-tuning tasks, we adopted full-parameter fine-tuning. All experiments were conducted on 4×A100 GPUs, and each experiment was repeated three times, with mean and standard deviation reported.

