# OpenReview forum: "Reshaping Activation Functions: A Framework for Activation Function Optimization Based on Mollification Theory"
_ICLR.cc/2026/Conference — ICLR 2026 Conference Withdrawn Submission_

### Official Review · Reviewer_fiv1 · 2025-10-27

**Soundness:** 2
**Presentation:** 2
**Contribution:** 1
**Rating:** 2
**Confidence:** 3

**Summary:**

The paper proposes a framework for "reshaping" activation functions by smoothing non-smooth functions such as ReLU via mollification theory. Using the bump function kernel, the authors derive a closed-form smoothed variant (termed S-ReLU) that is claimed to preserve the representational strengths of ReLU while providing better gradient stability and improved training.

**Strengths:**

- **Unified Framework**: The paper offers a unified theoretical framework for activation function optimization based on mollification theory. This approach systematically transforms non-smooth functions into smooth ones while aiming to retain their desirable representational properties.
- **Empirical Performance**: The experiments seem good, and the reported performance on several benchmarks (including challenging datasets like ImageNet-1K and tasks in LLM fine-tuning) suggests that S-ReLU can lead to favorable outcomes compared to existing activation functions.

**Weaknesses:**

1. **Lack of Strong Motivation:**
   While many smooth activation functions already exist—such as the sigmoid, GELU, SiLU, Mish, and Softplus—the paper does not convincingly justify the need for a unified framework to smooth activation functions. The motivation for developing an entirely new approach via mollification theory remains unclear compared to adopting or slightly modifying existing smooth activations.

2. **Limited Novelty in Theoretical Contribution:**
   The smoothing technique presented is essentially a standard method from classical analysis, where a sequence of smooth functions is constructed to approximate a target function. As such, most of the theoretical results are elementary and well-known in analysis, lacking significant novel insights or breakthroughs for the deep learning community.

3. **Narrow Analysis on the Role of Activation Smoothness:**
   The paper’s investigation into why smooth activation functions play a crucial role in the training process is limited. It primarily considers the Lipschitz constant as a measure of gradient stability, but the connection between the Lipschitz constant or other properties of activation functions and training performance is not thoroughly explored or justified. This leaves the reader with an incomplete understanding of how smoothing benefits training dynamics and overall model generalization.

**Questions:**

1. The paper suggests that to reduce the approximation error of the smoothed activation function, a small $\delta$ is required (Line 263). However, the analysis in Lines 313–318 indicates that a relatively large $\delta$ is needed to lower the Lipschitz constant of the smoothed function. This appears to create a trade-off where improving one aspect potentially worsens the other. How do the authors reconcile this conflict? Is there a clear strategy or optimal choice of $\delta$ that simultaneously guarantees a small approximation error and a reduced Lipschitz constant?

2. How sensitive is the proposed method to the choice of $\delta$ across different datasets and architectures? Is there evidence that a single $\delta$ (or a similarly derived strategy) generalizes well, or is extensive hyperparameter tuning required for each setting?

3. Does the imposed trade-off affect the network’s representational or expressive capacity? In other words, does achieving a lower Lipschitz constant at the expense of higher approximation error compromise the underlying ability of the network to capture complex functions?

**Details Of Ethics Concerns:**

No Concerns

---

### Official Review · Reviewer_eXbh · 2025-10-30

**Soundness:** 2
**Presentation:** 3
**Contribution:** 1
**Rating:** 2
**Confidence:** 4

**Summary:**

This paper presents a framework for smoothing non-smooth activation functions based on mollification theory. The framework aims to retain desirable properties and address issues of non-differentiability. Using the Epanechnikov kernel, the authors derive Smoothed ReLU (S-ReLU), a $C^2$-continuously differentiable and $1$-Lipschitz continuous activation function. Experimental results indicate that S-ReLU performance exceeds some existing ReLU variants in image classification tasks and large language model fine-tuning.

**Strengths:**

1. This work presents a mathematical framework based on mollification theory for smoothing activation functions, establishing a quantitative relationship between smoothness and gradient stability.

2. S-ReLU has $C^2$ continuous differentiability and a Lipschitz constant of 1.000, supporting stable gradient flow.

3. S-ReLU outperforms existing activation functions in some tasks, including image classification and large language model fine-tuning.

**Weaknesses:**

1. There is only one non-smooth point for ReLU, and the novelty of smoothing this single point is a small patch to this function, leading to insufficient impact to the ML community.

2. The smoothing technique is a well-investigated standard convolution calculation in variational analysis [1]. The approximating property is also well-known and straightforward because the smoothing kernel in Definition 1 has a compact support set and a finite integration. Such deductions are also demonstrated in textbooks on real analysis, functional analysis, and variational analysis, such as [1].

3. In the experiments, crucial baselines are missing. Sigmoid, ReLU, LeakyReLU, softplus [2], and swish function [3] should be included for comparisons. Compared with the only non-smooth point, the flat plot of ReLU-type function when $x<0$ is more detrimental to the training. LeakyReLU is proposed to fix this problem, so it should also be compared in the experiments.

4. $100$ epochs are not enough. At least $1000$ epochs should be implemented to ensure sufficient training. The performances of the current competitors seem rather too bad in Table 1 due to insufficient training.

5. Dropout and DropPath are crucial, standard regularization techniques for Vision Transformers (ViT, DeiT, TNT), essential for achieving state-of-the-art performance and reducing overfitting, especially on smaller data sets like CIFAR-100. Hence experiments with Dropout and DropPath settings should be conducted to further check the performance of S-ReLU.

[1] R. T. Rockafellar and R. J.-B. Wets, Variational Analysis. Springer Science \& Business Media, 2009, vol. 317.

[2] Xavier Glorot, Antoine Bordes, Yoshua Bengio Proceedings of the Fourteenth International Conference on Artificial Intelligence and Statistics, PMLR 15:315-323, 2011.

[3] Ramachandran, P., Zoph, B., \& Le, Q. V. (2017). Searching for activation functions. arXiv preprint arXiv:1710.05941.

The overall contributions of this paper appear to be rather limited to the ML community.

**Questions:**

See weaknesses.

---

### Official Review · Reviewer_j9N1 · 2025-10-31

**Soundness:** 2
**Presentation:** 1
**Contribution:** 2
**Rating:** 2
**Confidence:** 4

**Summary:**

This paper proposes a method to systematically create smooth activations functions by applying mollifiers on non-smooth activation functions. This is motivated by the fact that smooth variants of ReLU, such as GeLU or SiLU are widely used in modern architectures, and by theoretical studies that suggest that smooth activations can lead to more stable training dynamics. The paper first demonstrates that smooth activation functions via mollification are infinitely differentiable, and that they are faithful approximations of the original non-smooth activation function they were created from. Afterwards, using a particular mollifier function, the authors propose a new smooth activation function S-ReLU, which has a Lipschitz constant of 1. Experiments on various neural network architectures show that the proposed activation outperforms all other activations considered, with significant margins.

**Strengths:**

1. The idea of creating smooth activation functions using mollifiers is quite principled, compared to previous methods of handcrafted design and brute-force searching of the activation function space.

2. The proposed S-ReLU activation shows very strong performance across different model architectures and training tasks, and the performance margin is quite high.

**Weaknesses:**

1. While a large part of the paper devotes itself to the proving the $C^\infty$ property of their smoothed activation functions (Section 4.3), and that they are close approximations to the original function (Section 4.4), these are well-known results in the literature regarding mollifiers (also referred to as approximate identities).

For example, the definition of Smoothed Kernel function (section 4.2) as well as the derivations for the approximation properties (section 4.4) can be found in Section 1.2.4 of [1].

Likewise all results from 4.2~4.4 can also be found in Appendix C.4 of [2] as well.

In light of such state of the literature, I believe that it would be better to provide only proof sketches and citations in the main text and move the actual proofs to the appendix.

- [1] L. Grafakos, Classical Fourier Analysis, Third Edition. Springer. (2014).
- [2] L. C. Evans, Partial Differential Equations. American Mathematical Society. (1998)

2. Related to the previous point, the paper does not contain any references to the literature regarding mollifiers or approximate identities. The related works section also lacks discussion regarding this as well. I believe that this needs must be revised to make the contribution of the paper clearer.

3. It is not entirely clear from the paper why their method presents well-performing activation functions. The paper's brief discussion in section 4.1 applies to all smooth activation functions. The authors also discuss the Lipschitz constant of their activation function, but the Lipschitz constant of activations is nontrivially related to the Lipschitz constant of the entire neural network and it is the latter that is directly related to the performance of the neural network models.
I believe that either additional experimental or theoretical studies are needed to provide insights into why their method works so well.

**Questions:**

**Major**

1. Related to my comments in the weaknesses sections, is there a reason why the mollification approach gives activations that perform particularly well?

1-1. I am not sure if the Lipschitz constant of the proposed activations sufficiently explains the superior performance of the S-ReLU activation. Even in the references the authors cite (Zhou et al. (2019), Khromov & Singh (2024)) it is the Lipschitz constant of the network that is deemed important, not that of the activation. Note also that determining the Lipschitz constant of a neural network is quite non-trivial (Fazlyab et al. (2019)).
Can authors provide additional information about the importance of the Lipschitz constant of the activations?

1-2. Again, if the Lipschitz constant of activation function is a major factor for the model performance, why does ELU perform so badly in the experiments? Under the default settings of alpha=1.0, ELU should also have unit Lipschitz constant.

1-3. Note that for GeLU-like activations, one can trivially tune the Lipschitz constant to be 1 (or less than 1) by simple rescaling - see the LipSwish activation in (Chen et al, Residual Flows for Invertible Generative Modeling. NeurIPS (2019)).
Is there a performance gap between these scaled variants of GeLU, SiLU, etc and the proposed S-ReLU?

2. I am surprised by the large performance gap between S-ReLU and all other activations. What are the performance numbers like for ReLU? Does S-ReLU outperform this as well?
If the authors were to create another activation using a difference choice of the mollifier kernel, would it have similar performance improvements?

**Minor**

3. Is there a method to select the parameter $\delta$, or must this be treated as a hyperparameter and sweeped?

4. If the authors were to train the models further until the accuracy plateaus and the model fully converges, would the activations perform similarly, or would there still be a performance gap? In other words, does S-ReLU speed up the model convergence, but the final model performances are similar, or does it actually allow the model reach better final performance?
Providing the training curves may also help in this regard.

5. For the LLM finetuning, did the authors replace all activations of the model with S-ReLU, then finetune all parameters? Asking for the sake of clarity.

6. What is the per-epoch runtime of the models with different activations. Is S-ReLU particularly expensive to calculate?

7. In definition 1, the authors introduce a particular form of the smoothing kernel $\phi$, but do not use it in the rest of the paper. I assume the authors have mistakenly included the definition for the standard mollifier here. Is this correct?

---

### Official Review · Reviewer_Fdjw · 2025-10-31

**Soundness:** 2
**Presentation:** 2
**Contribution:** 2
**Rating:** 2
**Confidence:** 4

**Summary:**

The paper introduces S-ReLU, a smooth $\mathcal{C}^2$ version of ReLU obtained by convolving it with the Epanechnikov kernel. This yields a closed-form, tunable activation meant to improve gradient stability. The authors prove mollifier properties (smoothness, uniform approximation) and test S-ReLU on CNNs, Vision Transformers, and small LLMs, reporting somewhat modest accuracy gains over ReLU/GELU/SiLU.

**Strengths:**

- $S_1$: The paper presents a mathematically grounded construction of a smooth activation function, $S$-ReLU, derived by mollifying ReLU with the Epanechnikov kernel, yielding a closed-form $C^2$ function with a tunable parameter $\delta$.
- $S_2$: The theoretical analysis establishes smoothness and uniform approximation properties using standard mollifier results, providing a rigorous analytical foundation.
- $S_3$: The proposed activation is simple, analytically tractable, and could serve as a drop-in replacement for ReLU or GELU in modern architectures.
- $S_4$: The inclusion of a $\delta$-ablation highlights interpretability and control over the smoothness–performance trade-off.

**Weaknesses:**

- $W_1$: The theoretical argument connecting smoothness to gradient stability is only partially sound, explaining my "fair" grade in the soundness part. The Lipschitz-based derivation assumes a globally bounded activation input or $f \in L^{\infty}$, which is not stated. Without this, the “larger $\delta \Rightarrow$ more stable gradients” conclusion lacks formal validity and may hold only locally.

- $W_2$: The conceptual novelty is limited. I would not say this is a big weakness, but it is striking as the idea of smoothing ReLU to obtain differentiable or twice-differentiable activations is well studied (e.g., Softplus, ELU, CELU, Swish/SiLU, Mish). The use of an Epanechnikov kernel and the analytic closed form are elegant but do not constitute a fundamental advance beyond prior smooth activations.

- $W_3$: The experimental setup shows strong inconsistencies across datasets. CIFAR results display unrealistically low baselines for GELU and SiLU, leading to exaggerated relative improvements for $S$-ReLU. This makes me doubt about the choice of the training recipes used to build the baselines.

- $W_4$: The comparison set is incomplete. The paper omits several competitive smooth activations (Softplus, CELU, ELU, Mish) in key benchmarks, including LLM fine-tuning. As a result, it is unclear whether $S$-ReLU offers improvements over all established smooth alternatives or merely over specific baselines.

- $W_5$: The improved gradient stability is not empirically supported by diagnostics. There are no plots of gradient norms, activation distributions, or curvature proxies, making it impossible to verify the mechanism the theory aims to justify.

- $W_6$: Practical efficiency and deployment considerations are absent. I am not sure whether $S$-ReLU would be as cheap to compute as ReLU or SiLU. As it is a piecewise quartic polynomial it seems more expensive than ReLU or SiLU.

- $W_7$: Presentation quality is uneven. There are typos (“Poof” in Appendix B), notational errors (“SiLU” instead of “S-ReLU” in Remark 4), and unclear wording around extremum statements. The related work section could better contextualize prior smooth activations and clarify how the proposed approach differs theoretically and empirically.

**Questions:**

- $Q_1$: Can the authors restate the theoretical link between smoothness and gradient stability with explicit assumptions (could be bounded preactivations or $f \in L^{\infty}$?) and provide a formally valid proof or corrected bound supporting this claim?

- $Q_2$: Why was the Epanechnikov kernel specifically chosen for the mollification? Does its “optimality” in density estimation translate into measurable benefits for neural network optimization compared to Gaussian or other smooth kernels?

- $Q_3$: In the LLM experiments, why were only GELU baselines included? Would $S$-ReLU$'$s benefits persist when compared to SiLU or Swish, which are, to the best of my knowledge, standard in most transformer-based models?

---

### Note · Authors · 2025-11-15

I have read and agree with the venue's withdrawal policy on behalf of myself and my co-authors.